# Metabolomics in Parkinson’s Disease and Correlation with Disease State

**DOI:** 10.3390/metabo15030208

**Published:** 2025-03-18

**Authors:** Elena A. Ostrakhovitch, Kenjiro Ono, Tritia R. Yamasaki

**Affiliations:** 1Department of Neurology, University of Kentucky, Lexington, KY 40536, USA; eos226@uky.edu; 2Lexington VA Medical Center, Department of Neurology, Lexington, KY 40502, USA; 3Department of Neurology, Kanazawa University Graduate School of Medical Sciences, Kanazawa 920-8640, Japan; onoken@med.kanazawa-u.ac.jp

**Keywords:** Parkinson’s disease, metabolomics, glycolysis, TCA cycle, pentose phosphate pathway, kynurenine, fatty acid

## Abstract

Changes in the level of metabolites, small molecules that are intermediates produced by metabolism or catabolism, are associated with developing diseases. Metabolite signatures in body fluids such as plasma, cerebrospinal fluid, urine, and saliva are associated with Parkinson’s disease. Here, we discuss alteration of metabolites in the TCA cycle, pentose phosphate pathway, kynurenic network, and redox system. We also summarize the efforts of many research groups to differentiate between metabolite profiles that characterize PD motor progression and dyskinesia, gait and balance, and non-motor symptoms such as depression and cognitive decline. Understanding how changes in metabolites lead to progression in PD may allow for the identification of individuals at the earliest stage of the disease and the development of new therapeutic strategies.

## 1. Introduction

Parkinson’s disease (PD) is a progressive neurodegenerative disease characterized by motor and non-motor features. PD is the most common neurodegenerative movement disorder. With increased life expectancy, the prevalence and incidence of PD are expected to increase [1]. In 1990, 2.5 million individuals had Parkinson’s disease globally. The number increased to 6.1 million in 2016, with another doubling projected by 2040 [2,3]. PD affects 1–2% of individuals who are older than 60 years of age, with prevalence rate of 150/100,000 [4]. The diagnosis of PD relies on clinical symptoms, medical history, and response to dopaminergic therapy. DATscan (dopamine transporter scan) helps differentiate Parkinson’s from other movement disorders like essential tremor. However, it does not distinguish between different types of atypical parkinsonism. Mounting evidence suggests that a constellation of subtle clinical changes may precede motor manifestation by years or even decades [5]. Parkinson’s disease stems from the loss of dopaminergic neurons in the substantia nigra pars compacta. The loss of more than 30–50% of dopaminergic neurons is marked by symptoms like tremor, bradykinesia (slowness of movement), rigidity, and postural instability [6,7,8,9]. Pathogenic mechanisms involved in PD are only partially understood and include α-synuclein misfolding and aggregation, impaired protein clearance (autophagy-lysosomal system), calcium dysfunction, the endoplasmic reticulum (ER) stress initiated by protein misfolding, neuroinflammation, and oxidative stress [10]. The hallmark pathologic feature of most forms of PD is the presence of neuronal inclusions of aggregated α-synuclein in the form of Lewy bodies along with nigrostriatal degeneration. A lack of reliable and objective measures for the detection, diagnosis, and progression of Parkinson’s disease is a major stumbling block. There is also a need for identification of individuals at the earliest stages of the disease process, before symptoms develop, to optimize therapeutic intervention.

Although PD is not the result of a metabolic disorder, disruption of cellular energetics and metabolic changes play an important role in the development of PD. The presence of metabolic syndrome components (hypertension, dyslipidemia, and diabetes) increases the risk of PD [11,12]. Metabolomics allows for the comprehensive identification and quantitation of substrates and products of metabolism (the metabolome) and is considered an important tool for identifying metabolic biomarkers of the disease state. Many factors, including age, gender, diet, drug exposure, and the presence of the disease, influence the metabolome. Recent technological advances have allowed for the characterization of hundreds of metabolites. Comprehensive metabolic profiling of biofluids (serum/plasma, CSF, urine, saliva) is a noninvasive tool that may be harnessed to help predict disease incidence, severity, and progression. The advantage of metabolomic analysis lies in its ability to discriminate more than 6500 small molecule metabolites that define the chemical signature of biological activity and to identify qualitative and quantitative changes [13].

Metabolomic methods are varied in their approach and results. These include 1H nuclear magnetic resonance spectroscopy (NMR)- and mass spectrometry (MS)-based approaches. High-field magnetic resonance spectroscopy (MRS) uses stable isotopes such as ^13^C and ^1^H to analyze the metabolic flux in vivo under physiological and pathological conditions [14]. NMR has high-throughput capability and low cost per sample but relatively limited detection of low molecular weight molecules [15]. MS-based metabolomic analysis has a higher capability to detect a broad range of metabolites. The specificity and sensitivity are enhanced by mass spectrometric high-resolution separation techniques, particularly gas chromatography/mass spectrometry (GC/MS), liquid chromatography/mass spectrometry (LC/MS), or capillary electrophoresis/mass spectrometry (CE/MS). The use of LC-MS/MS, LC-selected reaction monitoring (SRM)/MS, and LC-high-resolution MS (HRMS) provides an orthogonal dimension of chromatographic separation and structural information that are ideal for the analysis of complex analytes that are present in biological matrices [16]. GC/MS (in over 30 studies), LC/MS (in over 40 studies) and nuclear magnetic resonance are the most commonly used techniques to identify and quantify metabolites within plasma, urine, or cerebrospinal fluid [17].

Analysis of identity and changes in concentration of metabolites may be utilized to provide a better understanding of the biological changes during the development and progression of pathological conditions, including Parkinson’s disease. Metabolic biomarkers for diagnosis and prognosis of Parkinson’s disease have been reviewed in several papers [18,19,20,21,22,23,24]. In our review, we discuss differences in metabolomic profiles according to motor and non-motor symptoms.

## 2. Discussion

### 2.1. Metabolic Pathways Affected in Parkinson’s Disease

#### 2.1.1. Glycolysis, TCA Cycle, Galactose, and Mannose

Since ~85% of PD cases are not genetically determined, metabolomic analysis holds great promise for discovering clinically relevant biomarkers. Numerous publications indicate alterations in various metabolic pathways in the early, mid, and advanced stages of PD. MRI and position emission tomography (PET) imaging studies have documented glucose hypometabolism in the PD brain. Even at early stages of the disease, PD patients exhibit reduced glucose consumption, a hallmark of PD [25,26,27]. Reduction in cortical glucose metabolism has also been reported in newly diagnosed PD [28]. Reduced glucose consumption in the frontal lobes and the caudate/putamen may occur at early stages of PD [29]. [^18^F]-fluorodeoxyglucose (FDG) position emission tomography analysis has shown reduced cerebral blood flow and glucose hypometabolism in the premotor cortex and in parietal association regions in PD patients [26,30,31,32]. Characteristic glucose hypometabolism is shown with FDG-PET in posterior temporoparietal, occipital, and sometimes frontal areas, accompanied by glucose hypermetabolism in the putamen, sensorimotor cortex, and cerebellum [33]. In PD with three years of symptom duration, Teune et al. found metabolic reductions in the occipital pole, inferior parietal and prefrontal cortex [34].

There is some evidence for an association between PD and diabetes mellitus, and the prevalence of diabetes has been reported to be 10% [35]. Numerous studies have found an abnormal glucose tolerance in PD patients and an increased incidence of PD in type 2 diabetes mellitus (DM2) [36,37,38]. PD patients with diabetes presented with worsened motor symptoms and cognitive impairment [39,40]. In DM2, decreased total brain volume and lowered glucose metabolism were found in the frontal lobe, sensory motor cortex, striatum, and right orbital part of the inferior frontal gyrus [41,42]. It has also been reported that brain hypometabolism was associated with the prediabetic condition [43]. This evidence suggests that regional glucose hypometabolism may precede the full clinical manifestation of some diseases by years. Furthermore, the circulating ER unfolded protein response regulator Glucose-regulated protein 78/Binding immunoglobulin protein (GRP78/BiP) was identified as a biomarker of metabolic disease as well as PD. The level of CRP78/BiP predicts the development of PD up to 7 years before symptom onset, as well as symptom severity [44,45,46]. The corrected pre-PET analysis showed hypometabolism in the right superior frontal gyrus, precuneus/posterior cingulate gyrus, left posterior orbital gyrus, right calcarine cortex, and right orbital part of the inferior frontal gyrus in the diabetic group [42]. Insulin dysfunction reduces the release and clearance of dopamine, induces brain inflammation, and worsens oxidative stress [47].

PD-associated metabolic changes are seen in glycolysis, tricarboxylic acid (TCA), pentose phosphate pathway (PPP), and lipid and ketone body metabolism when compared to healthy individuals [48]. The expression of most glycolytic genes is downregulated with the exception of some of the upper glycolysis hexokinases (HK2 and HK3), phosphofructokinase (PFKL), and aldolase (ALDOB) (Figure 1) in the substantia nigra pars compacta of PD patients [49]. Suppression of glyceraldehyde 3-phosphate dehydrogenase (GAPDH), phosphoglycerate kinase, and pyruvate kinase creates an energy supply chain decrease and, finally, energy shortage (Figure 1). Zagare et al. recently reported that lactate dehydrogenase A, the key regulator of the final step of glycolysis, is nearly two-fold downregulated in idiopathic PD-derived midbrain neural precursor cells [50]. NMR spectroscopy analysis showed a significantly elevated ratio of lactate to N-acetylaspartate, from 0.05 in control subjects to 0.11 in patients with Parkinson disease (Table 1) [51]. The level of pyruvate was increased in drug-naïve PD patients (n = 43) compared to healthy controls (n = 37) [52]. Accumulation of lactate and pyruvate in plasma (Figure 1) support pyruvate dehydrogenase deficiency, dysfunction of TCA, and energy metabolism deficit.

Umhau et al. explored the relationship between blood glucose and cerebrospinal fluid metabolite concentrations using high-pressure liquid chromatography [69]. The authors found that peripheral blood glucose concentrations correlated with the increased cerebrospinal fluid concentrations of the dopamine metabolite, homovanillic acid (r = 0.37 *p* = 0.017, n = 41), and the noradrenaline metabolite 3-methoxy-4-hydroxyphenylglycol (r = 0.52 *p* = 0.001, n = 41). These correlations support the idea of a homeostatic relation between brain neurotransmitter activity and blood glucose. Another study performed an analysis of CSF lactate in 101 drug-naïve PD patients stratified by motor progression, as described by Hoehn and Yahr (H-Y) scale [53]. Liguori et al. demonstrated higher lactate levels in the more impaired H-Y Stage 3 compared to H-Y Stage 2, as well as higher levels in H-Y Stage 2 than in H-Y Stage 1 and controls (Table 1) [53]. CSF levels of lactate negatively correlated with dopamine concentration. Thus, the increased level of lactate in more advanced PD may reflect upregulation of the activity of the astrocyte-neuron lactate shuttle, which is known to play a major role in central nervous system homeostasis and energy metabolism. This may have a significant impact on brain function, including memory [70,71,72]. A mitochondrial proteome analysis demonstrated downregulation in the pyruvate dehydrogenase complex (PDH), TCA cycle (IDH3A, isocitrate dehydrogenase; aconitase; CS, citrate synthase; OGDH, 2 oxoglutarate dehydrogenase or αKGDH, α-ketoglutarate dehydrogenase (Figure 1), mitochondrial complexes I, II, III, and IV activity, and fatty acid catabolism in regions with mild and severe pathology (substantia nigra, late frontal cortex, and putamen of early/late PD) [73]. Moreover, regions with no pathology (early frontal) still exhibited alterations in the TCA cycle. Mitochondrial 2-oxoglutarate malate carrier exhibited an increase across several brain regions with moderate and mild pathology (putamen, parahippocampus, and cingulate cortex). These changes indicate altered metabolic flux and mitochondrial dysfunction across various brain regions in early PD brain.

Fructose 6-phosphate is the point of divergence between glycolysis, fructose, galactose, and mannose metabolic pathways (Figure 2). Although glucose is the preferred energy source for the brain, fructose and galactose can be funneled into the glycolytic pathway. Mannose, the C2 epimer of glucose, can also be converted into fructose 6-phosphate. First, mannose is phosphorylated by hexokinase to generate mannose-6-phosphate (with kinetic properties similar to glucose phosphorylation). Then, mannose is converted into fructose 6-phosphate, which enters glycolytic pathway (Figure 2). Willkommen et al. reported that the D-glucose-6-sulfate and α-mannosylglycerate, which are part of fructose and mannose metabolism, were increased by 40% in the cerebrospinal fluid of PD patients [74]. Excess glucose due to glycolytic suppression may be converted into sorbitol. Indeed, an increase in hexose alcohols, including sorbitol, galactitol, and mannitol (Table 1), was detected in CSF in PD using multiplatform MS (LC-MS, GC-MS, LC-MS ESI) screening analysis [65].

#### 2.1.2. Pentose Phosphate Pathway

In PD, there is also evidence for alteration of the pentose phosphate pathway (PPP), which is also referred to as the phosphogluconate pathway or hexose monophosphate shunt. Glycolysis provides the 6-carbon glucose-6-phosphate utilized in the PPP. There are two phases in the PPP: oxidative and non-oxidative (Figure 3). NADPH is produced in the first oxidative phase. In the second, non-oxidative phase, 5-carbon sugars, including ribose-5-phosphate and erythrose-4 phosphate, are generated. NADPH is used in reductive biosynthesis, such as fatty acid synthesis, ribose biogenesis, and oxidative defense [75]. Ribose-5-phosphate is used to synthesize nucleotides, and erythrose-4 phosphate is used to synthesize aromatic amino acids. Neurons use glucose via the PPP to maintain their antioxidant status at the expense of their bioenergetic purposes to generate ATP (Figure 3). PPP activity accounts for approximately 5% of glucose metabolism in cortical neurons, 4% in cerebellar neurons, and 3% to 5% in astrocytes [76]. Reduced glucose availability affects the shunting of hexoses through the pentose phosphate pathway, a main route for converting hexoses into pentoses and producing reducing equivalents in the form of NADPH.

Levels of NADPH-producing enzyme glucose-6-phosphate dehydrogenase, the rate-limiting enzyme of PPP, and 6-phosphogluconate dehydrogenase were reduced in the putamen of early-stage PD and the cerebellum of early- and late-stage PD (Figure 3) [77]. It was also shown that the level of transketolase (TKT), a key enzyme in the PPP catalyzing the reversible reaction of D-xylulose 5-phosphate and D-ribose 5-phosphate to form D-glyceraldehyde 3-phosphate, is decreased in the substantia nigra of PD patients [78]. Consistent with these data, the level of sedoheptulose, an intermediate in the non-oxidative portion of the pentose phosphate pathway, was decreased in the CSF of PD patients [74,77]. In the early stages of PD, lower levels of PPP enzymes in the putamen could lead to reduced levels of NADPH and increased oxidative stress. However, another study found that in advanced stages of PD, the level of NADPH and activity of glucose-6-phosphate dehydrogenase were increased by 2-fold in the cortex and 1.5-fold in putamen brain tissue [77]. Increased activity of glucose-6-phosphate dehydrogenase (G6PD) and the corresponding increase in the level of NADPH in the cortex and putamen brain tissue may reflect microglia activation, since activation of G6PD was positively associated with microglial activation [77,79,80]. Microglia with high expression of G6PD also produce excessive NADPH and high amount of ROS [79]. Unfortunately, neither NADPH nor G6PD have been examined in the substantia nigra of PD patients. In animal PD models, both microglial activation and loss of nigral DA neurons had a positive correlation with increases in the expression/activity of G6PD and the production of NADPH [79]. Elevated G6PD activity produces excessive NADPH and provides an abundant substrate to NADPH oxidase (NOX2), leading to the production of excessive reactive oxygen species (ROS) [79].

In plasma from PD patients, near-infrared (NIR) analysis showed an increase in alcoholic (R-OH) functional groups and decreased hydrocarbon (C-H) and nitrogenous (N-H) groups, consistent with the enhanced oxidative stress [81,82]. PD patients were differentiated from control subjects with a sensitivity of 74% and specificity of 72% [81]. However, this technique is not feasible for use in clinical practice due to a high degree of overlap between PD and control groups. Multivariate statistical comparison of NMR-detected metabolites in plasma between the PD patient group and control group showed reduced concentrations of formic acid and increased concentrations of succinate in PD (Table 1, Figure 1) [62]. Accumulation of circulating succinate has been described as a potent driver of inflammation and an indicator of mitochondrial dysfunction and perturbations in the citric acid cycle [47]. Indeed, elevated serum levels of proinflammatory interleukins correlate with the severity of PD [83,84,85]. IL-17A correlated with non-motor symptoms (NMS) scores, while IL-6 positively correlated with motor scores [83]. In a study looking at serum from 642 PD patients and 277 controls, TCA cycle remodeling was associated with a change in energy production, which was characterized by the accumulation of the proinflammatory metabolite succinate, decreased concentration of the anti-inflammatory itaconate, and increased cysteine-S-sulfate (Table 1) [67]. Itaconate and cysteine-S-sulfate were also associated with motor symptoms among patients. The level of cysteine-S-Sulfate increased two-fold with the progression of PD [68].

#### 2.1.3. Redox Metabolites

Gas chromatography time-of-flight mass spectrometry (GC-TOFMS)-based metabolomics analysis of 25 patients with diagnosed idiopathic PD and 12 controls identified six significantly altered plasma metabolites, including L-3-methoxy tyrosine, aconitic acid, L-methionine, 13-docosenamide, hippuric acid, and 9,12-octadecadienoic acid (alpha-linoleic acid), which discriminate PD from controls with an accuracy of 92% [20]. Liquid chromatography high-resolution mass spectrometry (LC-HRMS) detected several PD serum metabolites, including anti-inflammatory itaconate and the NMDA glutamatergic receptor agonist cysteine-S-sulfate [67]. PD patients had lower levels of itaconate and higher levels of cysteine-S-sulfate compared to controls (Table 1)). The integrated omics data revealed two-fold (log2FC) downregulation for propanoate, cysteine, and methionine metabolism [50].

Mitochondrial dysfunction and neuroinflammation contribute to the increased oxidative stress and degeneration of dopaminergic neurons in PD. Glutathione is one of the most abundant intracellular antioxidants. Glutathione is the ubiquitous tripeptide, L-γ-glutamyl-L-cysteinylglycine, that protects cellular constituents from damage caused by free radicals and peroxides. Studies looking at the antioxidant glutathione show good correlation between levels in brain tissue and peripheral biofluids which are consistent with widespread oxidative stress in PD. Several studies show decreased levels of glutathione (GSH) in postmortem brain tissue substantia nigra and CSF of patients with PD [60,86,87,88,89]. In postmortem brain tissue substantia nigra and CSF of 48 patients with PD, oxidized glutathione levels decreased by almost one-half (*p* = 0.005124) compared to controls [60]. Total glutathione levels are decreased by 40–50% in the substantia nigra and nigral dopamine neurons in PD patients compared to healthy subjects [87,90]. The plasma level of GSH is also remarkably lower in PD patients compared to healthy controls (22.2 ± 19.0 vs. 79.3 ± 63.6 µmol/L) [91]. Depletion of GSH was reported to correlate with the severity of disease [91].

Decreased plasma concentrations of cysteine and cysteinyl-glycine are also biomarkers for oxidative stress. The reduction in cysteine and, specifically, cysteinyl-glycine reflects an elevated GSH synthesis which may by altered by medications used to treat PD. Cysteinyl-glycine concentrations were decreased in the blood of PD patients when tested before and 60 min after a 50 mg/200 mg dose of carbidopa/levodopa and a 50 mg/150 mg/200 mg dose of carbidopa/levodopa/entacapone [92,93,94]. A strong inverse relationship was found between the level of cysteinyl-glycine and levodopa derivative, 3-O-methyldopa (3-OMD). Alterations of uric acid and glutathione metabolites in the plasma of PD patients support the presence of oxidative stress and depletion of glutathione [95,96]. The reduction in uric acid and increase in the level of the uric acid precursor, hypoxanthine, indicate an imbalance in purine metabolism, contributing to oxidative stress. Xanthine oxidoreductase catalyzes the oxidation of hypoxanthine to xanthine and xanthine to uric acid with ROS production. Xanthine oxidoreductase is constitutively an NAD+ dependent dehydrogenase, which can be transformed through the oxidation of two cysteine residues or through a partial proteolysis of the fragment containing cysteine groups [97,98]. Elevated homocysteine levels (another marker of oxidative stress) were observed in the plasma of PD patients, particularly those under levodopa treatment [99,100]. However, this did not seem to be driven by a medication effect, as there were no changes in plasma homocysteine in patients with restless leg syndrome without or with levodopa treatment [100].

A metabolite of the glutathione cycle, pyroglutamic acid, may function in glutamate storage and act to oppose the action of glutamate. Pyroglutamate was found to be increased in the CSF of PD patients [58,65]. Increased plasma levels of pyroglutamate and 2-oxoisocaproate (ketoleucine) may indicate increased oxidative and metabolic stress in PD patients [58]. One large PD patient cohort, the DeNoPa Cohort, showed a significant reduction in CSF dehydroascorbic acid, a form of vitamin C that enables brain transport and reduces oxidative stress [64]. The reduction in dehydroascorbic acid also indicates brain cell damage due to increased oxidative stress [64]. Compared to controls, a significant increase was observed in fructose, mannose, and threonic acid in the CSF of PD patients [64]. The high fructose causes an ATP depletion that triggers an inflammatory response and oxidative stress with resultant perturbation of functions of tissues [101].

The levels of α-ketoglutarate and pyruvate, part of the TCA cycle (Figure 1), are increased in blood serum and CSF of PD patients [52,74,102]. Glycerol-3-phosphate (G3P) had the significant upregulation in PD samples. G3P is an intermediate metabolite in several metabolic pathways, including NAD metabolism, glycolysis, and lipid metabolism. Recent targeted metabolomic analyses revealed changes in the TCA cycle in PD brain frontal cortex and putamen [103]. NAD metabolism integrated into energy metabolism is also impaired [22,104]. Analysis of NAD+ showed a decrease in NAD+ and NAD/NADH ratio [22,105,106,107].

Nicotinamide, the active form of niacin, is the precursor of NADH, which is indispensable for complex I function. The increase in the level of nicotinamide was shown to be associated with the development of PD [108]. The excess of released nicotinamide may be methylated to 1-methylnicotinamide (MNA) in the brain. A superoxide generated by MNA via complex I inactivates complex I subunits. Niacin deficiencies are common in PD [109]. Niacin deficiency is also exacerbated in PD by dopaminergic medications [110]. It was shown that niacin supplementation reduces the risk of PD as well as improves rigidity and bradykinesia in patients with PD [107,111,112,113].

Decreased concentration of NAD metabolites is associated with brain adenosine triphosphate (ATP) levels and phospholipid-related metabolites [114,115]. The increased damage of complex I in mitochondria isolated from the frontal cortices of PD patients was attributed to the inhibition of NADH-stimulated electron flow [116]. The decrease in the NAD pool can result from decreased synthesis and increased consumption. The NADH imbalance results in mitochondrial dysfunction, impaired glucose metabolism, and oxidative stress. NAD+ is synthesized via de novo biosynthesis from tryptophan via the kynurenic pathway, nicotinic acid, and the salvage pathways (Figure 4). The first step of NAD salvage is catalyzed by nicotinamide phosphoribosyltransferase (NAMPT), an adipokine that plays a role in lipid and glucose metabolism. NAMPT consists of intracellular (iNAMPT) and extracellular NAMPT (eNAMPT). iNAMPT is vital in maintaining energy production via the homeostasis of NAD, whereas eNAMPT controls functional tissue homeostasis, neural activation, and the release of inflammatory cytokines [117,118]. eNAMPT is upregulated in the plasma of early-stage and drug-naïve PD patients [119].

#### 2.1.4. Tryptophan and Kynurenine Metabolic Network

There is ample evidence for dysregulation of the tryptophan and kynurenin pathways in PD. Metabolomic studies of blood and CSF revealed severe downregulation of tryptophan metabolism associated with tryptophan catabolism [58,60,120,121]. A significant reduction in tryptophan was reported in PD CSF [58] and in PD serum (Table 2) [120]. The ratio of 3-hydroxykynurenine and kynurenic acid concentrations increased by 64% (*p* = 0.000835) in PD (n = 48) [60]. Serotonin (5-hydroxytrypamine or 5-HT) is a metabolite on the tryptophan pathway (Figure 4). PD is characterized by the loss of terminals in serotonin-containing neurons and serotonergic dysfunction is already evident at early stages of PD [122]. PD participants exhibit serotonin and serotonin transporter deficits in the caudate, middle frontal gyrus, inferior parietal lobule, and visual association cortex, as assessed by high-performance liquid chromatography analysis of PD brain tissue [123]. Low serotonin concentrations in plasma, CSF, and several regions of the brain were reported in PD patients [124,125,126,127,128]. PD patients (n = 82) presented significantly lower plasma levels of serotonin and its metabolite, 5-hydroxyindoleacetic acid, in PD (Table 2) [127].

The oxidative cleavage of tryptophan into kynurenine metabolites occurs in the kynurenine pathway (Figure 4). The kynurenine/tryptophan ratio, measured in blood, was found to be associated with frailty and reduced cognitive performance in PD [129,130]. The kynurenine pathway intersects with the de novo NAD+ pathway (Figure 4). Changes in the kynurenic pathway contribute to PD-related NAD reduction. Studies have found lower levels of kynurenic acid and a reduced ratio of kynurenic acid/kynurenine, higher quinolinic acid level, and the altered ratio of quinolinic acid/kynurenic acid in the plasma and CSF of PD patients [121,131]. At an advanced stage, PD patients demonstrated significantly lower levels of kynurenic acid, with a shift toward quinolinic acid compared to those at the early stage [121,131]. Changes in hydroxytryptophan and kynurenine accurately distinguished the early, mid, and advanced stages of PD patients from control subjects [132]. PD patients demonstrate a two-fold increase in the plasma level of a downstream metabolite of kynurenine, 3-hydroxykynurenine [133].

The tryptophan metabolite 3-hydroxykynurenine is involved in the generation of free radicals and enhanced oxidative stress. An increased level of 3-hydroxykynurenin was found in the putamen, prefrontal cortex, and pars compacta of the substantia nigra regions in postmortem PD brain tissue as well as CSF [134,135]. Mass spectroscopy analysis of CSF showed a significant increase in the 3-hydroxykynurenine in PD [58,60]. Plasma 3-hydroxykynurenine was shown to be associated with the severity of the disease, suggesting that it might serve as a marker of PD severity/progression [133].

Quinolinic acid is another metabolite of the tryptophan pathway which has been studied in PD (Figure 4). Quinolinic acid acts as a pro-inflammatory mediator, and the increase in its concentration in the brain is closely linked to inflammatory response. Enhanced levels of quinolinic acid in CSF correlated with the levels of acute-phase proteins, which are typically elevated in inflammatory conditions [133]. 3-hydroxyanthranilic acid/anthranilic acid ratios have been associated with inflammation [136]. Given the consistency of alteration seen in the kynurenic and tryptophan pathways in PD and correlation with specific disease features, this metabolic pathway may have relevance as a biomarker of the disease state in PD.

#### 2.1.5. Amino Acids

There have been multiple studies that have investigated amino acid metabolism in PD. NMR analysis showed upregulation of isoleucine, valine, alanine, glutamate, and glutamine in the plasma of PD patients [59]. Multiplatform MS analysis (LC-MS, GC-MS, LC-MS ESI) of CSF has also demonstrated a significant increase in isoleucine and ketoleucine in PD patients compared with healthy controls [65]. The increased levels of proline and its precursor, non-essential amino acid ornithine, were observed in the plasma and urine of PD patients [19,61,63,65,120,137,138]. Ornithine levels were also increased (fold = 1.19, *p* = 0.024) in the CSF of PD patients (n = 22) as compared to control subjects (n = 28) [65]. The highest multivariant analysis coefficient was for L-ornithine in the CSF of PD subjects with persistence of PD symptoms for at least 5 years chosen from participants in the BioFind study (clinical trial NCT01705327) [63]. In plasma, proline was found to be significantly increased (7.03%RSD) in PD compared to matched controls [65].

The catabolism of branch-chain amino acids (BCAAs), such as valine, leucine, and isoleucine, is linked to the TCA cycle, gluconeogenesis, lipogenesis, and ketone body production. The uptake of amino acids into the brain is rapid, with leucine influx into the parietal cortex being the fastest. Leucine competes with kynurenine for transport to the brain through blood–brain barrier [139]. Perturbations in the level of BCAAs and their catabolism influence excitation/inhibition balance in the brain and thus affect the entire function of the central nervous system. Recent evidence strongly associates the upregulation of isoleucine and valine in the blood of patients with PD [59]. Decreased branch-chain amino acid dehydrogenase complex activity controlling BCAA catabolism is likely the main cause of increased BCAA levels [59]. However, this may be a late finding in PD, as a recent study found no association between plasma levels of BCAAs and PD among subjects for whom blood was collected less than 60 months prior to diagnosis or shortly after [140].

#### 2.1.6. Lipid Pathways

Several lipid pathways, including fatty acid, bile acid metabolism, and the sphingolipid pathway are impaired in PD [62,141,142,143,144,145,146]. Many studies support a downregulation of free fatty acids (FFAs) and lipid metabolism-related pathways in PD [50,58,147,148]. PD patients have decreased serum concentrations of polyunsaturated fatty acids (PUFA), such as alpha-linolenic acid and linoleic acid, as compared to controls [148]. Decreased levels of oleic acid and increased levels of arachidonic acid were reported in PD [148,149,150]. The increased level of arachidonic acid in CSF was associated with increased oxidative stress and neuroinflammation [74]. The presence of arachidonic and docosahexaenoic acids may accelerate the aggregation of a-Syn [151]. Previously, perturbation of the sphingolipid pathway was also linked to the aggregation of a-Syn [145,149,150]. The decreased levels of glycosphingolipids, the product in the sphingolipid pathway, have been found in the plasma of PD patients [150].

It was reported that the development of PD is associated with the high plasma concentrations of short-chain fatty acids (SCFA), including propionate (C3) and butyrate (C4) and long-chain fatty acids, such as hexadecanoic acid and octadecanoic acid [152,153,154]. Reductions in fecal short-chain fatty acids and their increase in plasma corelated to specific gut microbiota changes and the clinical severity of PD [154]. Among other fatty acids, stearic acid, oleic acid, and palmitic acid showed a significant decrease in subjects with PD compared to control subjects [19,121,147]. Changes in fatty acids are associated with mitochondrial dysfunction and with oxidative stress (Figure 1). Analysis of long-chain fatty acids revealed a decrease in the levels of eicosapentaenoic, oleic, linoleic, octadecanoic acid, heptadecanoic, and palmitoleic acids in the plasma of PD patients [58]. The ratio of plasma levels of 2-oxoisocaproic acid to CSF levels of 3-hydroxyisovaleric acid was used to differentiate a subset of PD from controls. Medium- and long-chain fatty acids, including 5-dodecanoate, 3-hydroxydecanoate, docosadienoate, docosatrienoate, and decanoic acid, were increased in plasma and CSF of PD patients [60,74]. Higher intake of omega-3 PUFA has been associated with a reduced risk of PD [155,156,157]. The neuroprotective action of docosahexaenoic acid is thought to occur through modulation of microglial activity, increased striatal dopamine concentrations and increased synaptogenesis down-regulation of COX-2 expression, and decreased neuroinflammation [158,159,160,161]. Thus, studies on lipidomics may contribute to the development of effective treatments if the neuroprotective action of specific lipid profiles can be replicated by disease-modifying therapies.

A two-fold increase in the plasma levels of diacylglycerol (DAGs), the precursors of glycerophospholipid, phosphatidylcholine (PtdC), lysophosphatidylcholine (LPC), and lysophosphatidylethanolamine has been identified in PD [162,163]. Dopamine agonist treatment in PD is positively associated with an increase in the PtdC/LPC ratio. However, the Hoehn and Yahr scale and the disease duration did not influence lipid levels. DAGs with both monounsaturated and polyunsaturated fatty acid side chains (34:1; 36:1; 36:2; 36:4; 38:6) were significantly (from two- to four-fold) increased in the PD frontal cortex [164]. Frontal cortex levels of phosphatidylserines (PtdS 36:1, 36:2, 38:3) and phosphatidylglycerol (PtdG 32:0) were increased, whereas phosphatidylcholines (PtdC 34:5; 36:5; 38:5; 38:6) and lysophosphatidylglycerol (LPG 16:0) were decreased by up to two-fold.

#### 2.1.7. Gut Microbiota Metabolites

Gut microbiota disturbance (gut dysbiosis) is an emerging biomarker in PD [165]. Gut metabolites include aromatic amino acids, short-chain fatty acids, and metabolites derived from bile acids and cholesterol. Microbial dysbiosis is accountable for an upregulation of bacteria responsible for secondary bile acid synthesis in people with PD [166,167]. The depletion of short-chain FFA-producing bacteria has also been linked with gut dysbiosis in PD [168,169]. The increased gut permeability allows short-chain fatty acids to leak into blood circulation. The changes in fecal and plasma short-chain fatty acids could discriminate patients with PD from healthy control subjects and were associated with severity of the disease. SCFAs influence the release of serotonin (5-HT) and GABA. Changes in in the microbiota SCFAs were linked to the severity of PD [154,170]. The concentrations of short-chain acids, acetic (C2), propionic (C3), butyric (C4), branch-chain isobutyric (C4), and branch-chain saturated isovaleric (C5) acids were reduced in feces, whereas plasma concentrations were increased in patients with PD [170]. A few studies identified the increased plasma level of unconjugated bile acids and secondary bile acids, such as lithocholic acid and deoxycholic acid, in PD [96,167,171,172].

In addition, the salvage and de novo pathways of vitamin B12 biosynthesis were significantly enriched in the control group but not in the PD group. It was suggested that gut dysbiosis precedes the onset of PD’s movement symptoms and other early signs. The microbial-related p-cresol metabolites p-cresol sulfate and p-cresol glucuronide were found at high intensities in the serum of PD patients [67]. P-cresol and two of its metabolites, p-cresol sulfate and p-cresol glucuronide, were also found at higher intensity among the PD patients relative to controls (p-cresol log2FC = 0.41, 95% CI = 0.21, 0.60, FDR = 2.8 × 10^−3^). The levels of microbial-related p-cresol metabolites were correlated with age among PD patients [67].

Recent studies show a decrease in cholesterol levels and an increase in the unconjugated bile acids, cholic acid, deoxycholic acid, and lithocholic acid in the plasma of PD patients [96]. The increase in unconjugated bile acids in plasma levels might be due to increased bacterial degradation of conjugated bile acids or inefficient removal of unconjugated bile acids from the peripheral circulation due to gut microbiota disturbance in PD.

### 2.2. Differences in Metabolomic Profiles According to Motor Symptoms

#### 2.2.1. Motor Progression and Dyskinesia

Numerous metabolic studies have been conducted to identify biomarkers and metabolic pathways involved in the development and progression of PD. However, very few studies describe metabolic changes in association with motor symptoms in PD. Diagnosis of PD is based on the presence of bradykinesia, rest tremor, rigidity, and supportive criteria such as the presence of levodopa-induced dyskinesia (Figure 5) [173,174,175,176]). Zhang et al. found additional hypometabolism in the caudate nucleus and an inferior parietal lobule in patients with postural instability compared to tremor-dominant patients [177]. Significant glucose hypometabolism was observed in the ventral striatum of akinetic-rigid PD patients in comparison with tremor-dominant patients in another study [178]. It is believed that akinesia is related to dopaminergic depletion [179]. However, pathological processes in PD are not only limited to the dopaminergic system. The pathology of PD involves non-dopaminergic neurotransmission, including serotonergic, norepinephrine, cholinergic, and glutamateric dysfunction [180]. Analysis of clinical rating scales and imaging of dopamine and serotonin transporters in patients with early Parkinson’s disease showed that the severity of rest tremor correlates strongly with the loss of serotonin transporter in the brainstem raphe nuclei rather than the loss of dopamine transporter in the striatum [181]. The depletion of serotonin (5-HT) in the plasma and CSF of PD patients is well described [124,125,126,127,128]. It was also shown that the concentration of 5-HT in CSF negatively correlated with Hoehn and Yahr’s stages, the severity of rigidity, akinesia, and gait freezing (Figure 5) [128].

Dyskinesia is a levodopa-induced motor side effect of treatment that occurs in a subset of PD patients. In PD patients that developed levodopa-induced dyskinesia, a three-fold increase in the 3-hydroxykynurenine/kynurenic acid ratio in plasma occurred [121]. PD patients who did not develop L-DOPA dyskinesia despite L-DOPA treatment had increased levels of kynurenic acid and anthranilic acid in plasma and CSF [60,121]. The authors observed significant changes in kynurenine metabolism in levodopa-induced dyskinesia in PD, differentiating this group from controls, but also from non-dyskinetic PD [121]. An enhanced 3-hydroxykykynurenine/kynurenine ratio in plasma and CSF may be indicative of a change in kynurenine metabolites in the basal ganglia at the synaptic level [121]. The rise of quinolinic acid levels and depletion of kynurenic acid in the CSF of PD patients were associated with motor symptoms [133], as mentioned above. Quinolinic acid is an agonist of N-methyl-d-aspartate receptor (NMDAR) and N-methyl-D-aspartate (NMDA) receptors have also been implicated in L-DOPA-induced dyskinesias in PD patients [133]. Studies postulate that low kynurenic acid levels allow glycine to further potentiate NMDAR function which may facilitate L-DOPA induced dyskinesia [182,183,184].

The Unified Parkinson’s Disease Rating scale (MDS-UPDRS) motor scores were shown to correlate with increased plasma propionate and reduced fecal levels of propionate, acetate, and butyrate in patients with PD [137,154]. The change in plasma phenylcarnitine concentrations was moderately correlated with UPDRS, whereas changes in plasma 1,3-dimethylurate, aspartylphenylalanine, and phenylcarnitine were the best predictors of UPDR motor score [185]. In CSF, the strong correlation with UPDRS was for changes in benzoate concentrations. Microbial-related p-cresol metabolite, p-cresol glucuronide, was also associated with a higher Hoehn Yahr (HY) stage among PD patients [67].

Lower serum triglyceride levels were significantly associated with higher MDS-UPDRS III total scores and gait/postural instability sub-scores [186]. It was suggested that serum triglyceride may be a potential predictive biomarker for motor performance in PD patients. However, it was also reported that triglycerides were negatively correlated with cognitive function in PD [187,188].

#### 2.2.2. Gait and Balance

Balance and gait are important factors in PD as falls can significantly increase morbidity and mortality. Cholinergic dysfunction has been studied in relation to changes in gait and balance in PD [189]. A reduced choline level in the cerebrospinal fluid of PD patients results from impaired choline transport and modified phospholipid content [190]. Studies showed that the level of acetylcholine decreases in cerebrospinal fluid in PD patients with the postural instability gait disturbance (PIGD) phenotype compared to PD patients with the tremor-predominant phenotype [191]. Decreased thalamic cholinergic activity, independent of dopaminergic integrity, increases postural sway [192]. In one randomized clinical trial, cholinergic augmentation with a reversible and noncompetitive cholinesterase inhibitor (donepezil, at 10 mg/day for six weeks) did not affect measures of static or dynamic balance in people with PD [193]. Similarly, cholinergic augmentation with low-dose rivastigmine (5 mg/day) did not affect gait speed and stride time variability [194]. However, in a randomized double-blind, phase 2 trial, a high dose of rivastigmine of 12 mg/day over 12 weeks improved step time variability for everyday walking and a simple dual task with phonemic verbal fluency (walking while naming words beginning with a single letter) [195].

#### 2.2.3. Dysphagia

PD patients over 63.5 years of age and with a daily levodopa equivalent dose >475 mg show an increased risk of pharyngeal dysphagia [196]. Dysphagia, or difficulty swallowing, was significantly correlated with UPDRS and modified Hoehn and Yahr scale scores, and is likely to occur with disease progression. Cognitive impairment at baseline was significantly related to dysphagia aggravation (*p* = 0.042). PD patients with dysphagia exhibited a significantly greater increase in anxiety scores, severity of falls, and severity of the disease [197,198]. Low plasma citric acid, isocitric acid, 3-hydroxybutyric acid, and 3-hydroxyisobutyric acid, as well as antioxidant vitamins C and b-carotene, were shown to be associated with coughing, which may co-occur with dysphagia [199,200].

#### 2.2.4. Restless Leg Syndrome

Restless leg syndrome (RLS) tends to co-occur with PD [201]. PD patients with RLS had higher stages and scores of motor symptoms, as well as non-motor depression, anxiety, sleep disorders, fatigue, and apathy. The levels of ferritin, dopamine (DA), and 5-hydroxytryptamine (5-HT) were decreased in PD-RLS patient CSF, and the RLS score was negatively correlated with dopamine and 5-HT levels [202]. A strong correlation was shown between decreased serum level long-chain (polyunsaturated) fatty acids (valeric, linoleic, arachidonic and eicosapentaenoic acids), gamma-glutamyl amino acids, glutamate, xanthine, serine, L-phenylalanine, and increased levels of inositol metabolites (scyllo-inositol, chiro-inositol), glycerol, glycerol 3-phosphate, and RLS in PD [148]. Recent data suggest a significant association between serum levels of vitamin D, parathormone (PTH), and related metabolites and leg restlessness in PD [203].

### 2.3. Differences in Metabolomic Profiles According to Non-Motor Symptoms

Non-motor symptoms of PD are significantly impairing and include sleep disturbance, autonomic dysfunction (low blood pressure and urogenital dysfunction), urinary incontinence (the loss of bladder control), constipation, dysphagia, fatigue, thermoregulation, mood disorders (anxiety and depression), and cognitive decline, (Figure 5) [9]. Oculomotor behavior changes in patients with Parkinson’s disease (PD) were also suggested as a promising source of prodromal disease markers [204,205].

#### 2.3.1. Prodromal Symptoms

The prodromal symptoms in patients with PD include constipation, nausea, and vomiting. The prevalence of gastrointestinal issues reflects importance of the microbial composition in gastrointestinal tract in development of PD [206]. The microbial metabolites include short-chain fatty acids (SCFAs), amino acid metabolites, and secondary bile acid metabolites. The SCFAs are the product of bacterial fermentation of complex polysaccharides [207,208]. The decrease in short-chain FFA results in the downregulation of regulatory T cells that increases the proinflammatory response, which could be linked to gastrointestinal symptoms in PD patients [169,209]. Prodromal PD patients demonstrated significant changes in microbial fatty acid metabolism and queuosine and archeosine synthesis pathways, implying that these metabolites might serve as a marker to identify PD in the early state, prior to motor symptom manifestation [210].

#### 2.3.2. Depression

Non-motor symptoms, such as major depressive disorders and cognitive decline, are associated with poorer prognosis in PD patients. The occurrence of major depressive disorder symptoms occurs in 35% or more PD patients [211]. Substantial numbers of PD patients have moderate-to-severe depressive symptoms [212,213]. On the other hand, numerous epidemiological studies suggest a strong correlation between major depressive disorders and the development of PD [214]. Transcriptomic analysis of blood from patients with PD and major depressive disorder identified NAMPT as the most significantly upregulated parameter [119]. The nicotinamide metabolism pathway was upregulated in both depression and PD. Serotonergic dysfunction was shown to correlate with non-motor symptoms such as depression and pain in PD (Figure 5). Serotonin is involved in the regulation of the sleep–wake cycle, mood and emotions, cognition, and concentration [127]. The levels of serotonin or 5-hydroxytryptamine (5-HT) and 5-hydroxyindoleacetic acid (5-HIAA) are significantly decreased in PD patients with depression compared to PD patients without depression [215]. It was suggested that plasma levels of 5-HT and its metabolite 5-hydroxyindoleacetic acid may be considered peripheral markers for depression in PD [127].

The involvement of the glutamatergic system in mood disorders is based on pre-clinical studies of NMDA antagonists [216]. However, further studies found that people with depression have lower levels of glutamate in the brain than healthy subjects [216]. Glutamate levels were significantly decreased in the CSF of PD patients [217,218]. Analysis of the relationship between plasma levels of glutamate and the severity of depression showed that plasma levels of glutamate, alanine, and L-serine were reflective of the severity of depression [219]. Similarly, a high level of glutamate in the serum of PD patients was negatively associated with the severity of depression in PD [220].

#### 2.3.3. Circadian Dysfunction and Sleep Disturbance in PD

Circadian dysfunction and sleep disturbances are common in PD. More than sixty percent of PD patients develop sleeping problems [221]. Patients with PD had a four-fold decrease in the amount of circulating melatonin [222]. The circadian rhythm is regulated by the circadian clock. There is reciprocal regulation between the circadian clock and metabolism [223]. Misalignment of the circadian rhythm exacerbates or causes metabolic disorders, including diabetes [223]. The findings of circadian biology from transcriptomic, epigenomic, proteomic, and metabolomic research revealed that the circadian clock governs daily changes in metabolic activity and, in turn, is functionally dependent on metabolic changes [223]. The circadian metabolome and transcriptome are re-programmed by nutritional challenges, exercise, diseases, and aging. Some crucial metabolites that directly affect the circadian clock are taurine, formate, citrate, 3-indoxyl sulphate, carnitine, 3-hydroxyisobutyrate, trimethylamine N-oxide (TMAO), and acetate, which exhibited increased levels in sleep-deprived subjects (Figure 5). Eight additional metabolites—dimethylamine, 4-DTA, creatinine, ascorbate, 2-hydroxyisobutyrate, allantoin, 4-DEA, and 4-hydroxyphenylacetate—showed decreased levels in sleep-deprived subjects [224]. TMAO is a modulator of cholesterol and sterol metabolism. Other metabolites include polyamines and NAD+ [225,226]. Sleep disorders are associated with increased deoxy sugar, 9-hexadecenoic acid, arachidonic acid, xanthine, and DOPAC, and decreased levels of choline, ghrelinIn, and NAD+ [227,228]. In PD, striatal dopamine regulates CLOCK/BMAL1 [229]. Changes in the serotonin pathway (5-hydroxytryptamine, 5-HT) play a critical role in modulating sleep, arousal, mood, and emotion [230].

In a recent study, fifty-six PD patients were enrolled with respect to motor symptoms, of whom 10 had an REM sleep behavior disorder (RBD) before motor symptoms (PD-RBDpre), 19 had an REM sleep behavior disorder after motor symptoms (PD-RBDpost), and 27 were PD-RBD- [231]. Univariate logistic regression analysis of semi-quantitative 123I-FP-CIT-DAT-SPECT imaging revealed a significant positive association between PD-RBDpre and mean caudate-specific binding ratios, as well as between PD-RBDpre and the putamen asymmetry index. The authors concluded that different patterns of striatal dopaminergic dysfunction exist based on RBD onset in PD, with a more symmetrical putaminal impairment and a more severe caudate involvement in PD patients in which RBD precedes motor onset. Exogenous levodopa may impair 5-HT function by inhibiting tryptophan hydroxylase [232]. Moreover, the conversion of levodopa into dopamine leads to depletion of 5-HT [233]. It was demonstrated that the uptake of tryptophan and kynurenine derived from the peripheral circulation into the brain enhances kynurenic acid production in sleep deprivation-induced central fatigue [234].

#### 2.3.4. Cognitive Decline

Cognitive impairment is a common feature of PD. Mild cognitive decline is evident from the early stages of PD. Up to 20% of newly diagnosed PD patients show signs of mild cognitive decline, and the percentage increases during the course of the disease [235,236]. PD patients develop cognitive impairment ranging from subjective cognitive decline through mild cognitive impairment (MCI) to Parkinson’s disease dementia (PDD) [237,238]. MCI is considered a transitional stage from normal cognition to dementia. It was reported that the transition rate to PDD is much higher in PD patients with MCI [237,238]. Understanding the mechanisms that drive the progression from MCI to PD is extremely important, as dementia is a significant cause of morbidity and mortality in PD. PD dementia (PDD) occurs in 17% of patients at 5 years after diagnosis, increasing to 50% at 10 years after diagnosis and over 80% within 20 years from the initial diagnosis [238,239]. Compared with cognitively normal PD, the PD-MCI group had reduced metabolism in the inferior parietal and posterior temporal regions [28]. In patients with PDD, glucose hypometabolism was reported in the visual cortex, posterior cingulate cortex, and the lateral parietal, lateral temporal, and lateral frontal binding areas [240,241]. The decreased levels of fatty acids and increased levels of monoglycerides and diglycerides (42:6; 42:5) in PD serum correlate with development of PDD [162]. Overall, 24 ceramides, 24 DGs, and 17 TGs were increased in PD patients progressing to PDD. In contrast, 105 lipids, including 16 phosphatidylcholines, 14 bis(monoacyl)glycerophosphates, and 14 phosphatidylserines were decreased in PDD serum. Alteration of the interaction of α-Syn with PtdS in PDD may facilitate protein aggregation, and therefore, may contribute to the neuronal dysfunction responsible for cognitive decline [242,243,244,245].

The risk of dementia in the elderly was associated for a long time with increased levels of homocysteine (hyperhomocysteinemia) [246,247]. PD patients with homocysteinemia are more likely develop depression and cognitive impairment (Figure 5) [248]. Homocysteine levels were independently correlated with MCI in PD patients [249]. Demented PD patients were also likely to have higher plasma levels of homocysteine than non-demented PD patients [250,251,252,253]. The total level of homocysteine is high in PD CSF [254]. Hyperhomocysteinemia reduces tyrosine hydroxylase (TH) activity, leading to the degeneration of dopaminergic neurons and the progression of PD [255,256]. Furthermore, hyperhomocysteinemia induces oxidative stress and mitochondrial dysfunction [257,258,259]. In PD, levodopa further increases the homocysteine level in plasma and CSF [252,260]. Recently, Kalecky et al. reported the elevation of homocysteine in the PD frontal cortex [103]. The authors provided evidence of the generation of homocysteine through levodopa metabolism by catechol-O-methyltransferase (COMT) and its involvement in dementia. The population study that included the Pacific Northwest Udall Center (PANUC) clinical cohort showed that PD patients with elevated plasma homocysteine had lower scores on the digit symbol test, the Hopkins Verbal Learning Task, the delayed recall test, and semantic verbal fluency [261]. In longitudinal trials (SURE-PD:NCT000833690, STEADY-PDIII: NCT02168842, SURE-PD3: NCT02642393), homocysteine levels increased in PD patients who began levodopa but were not taking B12-containing supplement [262]. In these trials, 3.7%, 3.6%, and 1.1% of participants had low B12 levels, which are associated with advanced PD and related neuropathy and cognitive impairment.

Gut dysbiosis leads to systemic and neuroinflammation and subsequently affects cognitive ability [188,263]. It was reported that PD is accompanied by an increase in plasma levels of unconjugated bile acids (cholic acid, deoxycholic acid and lithocholic acid) and purine base intermediary metabolites, particularly hypoxanthine [188,263]. A comprehensive metabolomic analysis of plasma from Parkinsonian patients highlighted the importance of bile acids and purine metabolism in the pathophysiology of this disease [96]. Targeted metabolic analysis revealed the accumulation of several bile acids (BAs), including the metabolic indicator of the glycine conjugation of deoxycholic acid (DCA) to form glycodeoxycholic acid (GDCA) in PD cortex [103]. The increased glycine conjugation of secondary bile acid GDCA was observed in the PD group with cognitive impairment. Furthermore, low levels of chenodeoxycholic acid, cholic acid, and ursodeoxycholic acid were shown to be significantly associated with PD-MCI [264].

Plasma methylglyoxal is elevated in PD [77,265]. Impaired glucose utilization and accumulation of methylglyoxal promote the formation of advanced glycation end products [266]. PD is associated with increased protein glycation, particularly of α-synuclein [267,268]. Glycation of α-synuclein potentiates its oligomerization and accumulation in PD brain tissue [269]. Accumulation of advanced glycation end products was implicated in cognitive decline [270]. Consequently, it is feasible to suggest that PD patients with high plasma levels of methylglyoxal are more likely to develop dementia. Cognitive decline is also linked to mitochondrial fatty acid oxidation process. Instead, phosphatidylcholines (PC), which are phospholipids that incorporate choline, cholesterol ester (CE), and genes related to endoplasmic reticulum stress and cell regulation, were associated with worse cognition in PD [270].

## 3. Conclusions

Metabolomic analysis is an emerging area of interest in Parkinson’s disease. There is good evidence for alterations in multiple metabolic pathways in PD and some correlation between specific pathways and progression in motor and non-motor features. Further investigations are required to attain a deeper comprehension of how specific pathway alterations may reflect the pathophysiology of PD and to what extent metabolite profiles in biofluids reflect the metabolic milieu of the underlying neurodegenerative state.

Metabolomic analyses and data processing present several challenges in the standardization of the results. Luo et al. conducted meta-analysis of metabolite reproducibility from 74 studies published from 2003 to 2022 and identified 928 metabolites that were significantly changed in PD, with only 190 metabolites replicated with the same changes in more than one study [17]. The authors found that 60 metabolites were exclusively increased, 54 exclusively decreased, and 76 inconsistently changed. In total, 39 studies utilized plasma, 22 studies used serum, 17 studies used cerebrospinal fluid, 17 studies used brain tissue, and 6 studies used urine or feces. Inconsistency in results may arise due drug treatment, disease stage, use of different techniques for sample processing and different metabolomic analysis methods, and changes in the gut microbiome caused by diet and drugs. Other challenges in metabolomic monitoring include difficulty in metabolite identification and the lack of a standardized reference database. Metabolomic profiles must be tested in large, well-characterized cohorts with standardized collection procedures, analysis techniques, and references to validate findings.

Identifying metabolites during the development and progression of PD can provide new insights into the disease’s mechanisms and identify new targets for intervention. For example, a phase 1 clinical trial found that a high dose of nicotinamide riboside (NR), a form of vitamin B3, helps increase NAD+ levels in people with PD. Oral NR increases brain NAD levels and is associated with clinical improvement [271]. Specific diets may restore the gut microbiome and alter the clinical progression of PD. In clinical trials, the investigators plan to target the microbiome through diet or rifaximin, an antibiotic used to treat irritable bowel syndrome with diarrhea.

New therapeutic approaches are considered by targeting lactate and pH. For instance, topiramate and levetiracetam reduce the intracellular pH of hippocampal neurons and thus may be considered for PD treatment. Diabetes and obesity are common features of PD. Glucagon-like peptide (GLP-1) receptor agonists (GLP-1 RAs) are agents licensed for treating type 2 diabetes. GLP-1 RA showed efficacy in PD motor symptoms [272]. There are a few phase 2 trials that evaluate the treatment effect of semaglutide (NCT03659682), liraglutide (NCT02953665), and lixisenatide (NCT03439943) on motor symptom progression, as well as the MRI-based (NCT03456687) and FDG-PET based (NCT04305002) imaging markers of disease progression with exenatide [272]. The enhancement of the methylglyoxal scavenging system may provide new therapeutic opportunities to reduce the pathophysiological changes associated with carbonyl stress. The inhibitors of xanthine oxidoreductase can be beneficial in the treatment of PD. The xanthine oxidase inhibitor allopurinol has been shown to normalize endothelial dysfunction in individuals with Type 2 diabetes and reduce lipid peroxidation and the formation of reactive oxygen species [273].

Although research on dietary interventions in PD is just beginning, the results of the various ongoing nutritional trials suggest a beneficial effect of omega-3 fatty acids, vitamin D, B vitamins, and coenzyme Q supplementation for managing PD. The ketogenic diet improves mitochondrial metabolism, neurotransmitter function, and oxidative stress/inflammation. A 24-week ketogenic diet with restriction of protein and carbohydrates to 30% of the daily energy expenditure positively influenced gait and mobility, self-care, socialization, depression, and anxiety in PD adults [274]. The relationship between the gut microbiome and PD development suggests that probiotics, synbiotics, and vitamin B supplements may help with Parkinson’s disease (PD) symptoms by improving gut health and reducing inflammation.

Overall, metabolomics holds promise as an area with high potential for dissecting underlying pathophysiologic derangement in PD and identifying new biomarkers of the disease state.

## Figures and Tables

**Figure 1 metabolites-15-00208-f001:**
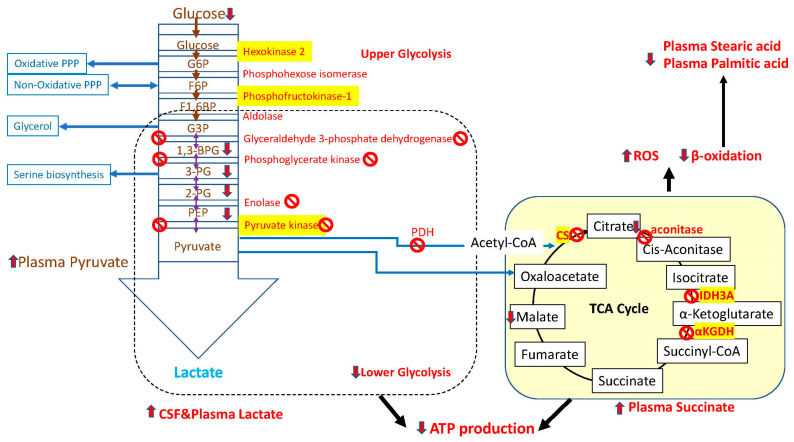
Schematic representation of glucose metabolism impairment and energy supply impairment in Parkinson’s disease and altered metabolites in plasma and CSF. The enzymes involved in each node are written in red and the regulatory enzymes are highlighted in yellow. Blue lines from different nodes of the glycolytic pathway represent the precursor molecules, which are shuttled between different pathways. The regulation of the enzymes is represented by the regulatory molecule and its effect is represented by an arrow. The inhibition of the enzymes is represented by a crossed red circle. The excessive accumulation of G6P leads to the inhibition of hexokinase 2. ATP, adenosine triphosphate; G6P, glucose 6-phosphate; F6P, fructose 6-phosphate; F1,6BP, fructose-1,6-bisphosphate; G3P, glyceraldehyde 3-phosphate; 1,3-BPG, 1,3- bisphosphoglycerate; 3-PG, 3-phosphoglycerate; 2-PG, 2-phosphoglycerate; PEP, phosphoenolpyruvate; TCA, tricarboxylic acid cycle; CS, citrate synthase; IDH3A, isocitrate dehydrogenase (NAD+) 3 catalytic subunit alpha; αKGDH, α-ketoglutarate dehydrogenase.

**Figure 2 metabolites-15-00208-f002:**
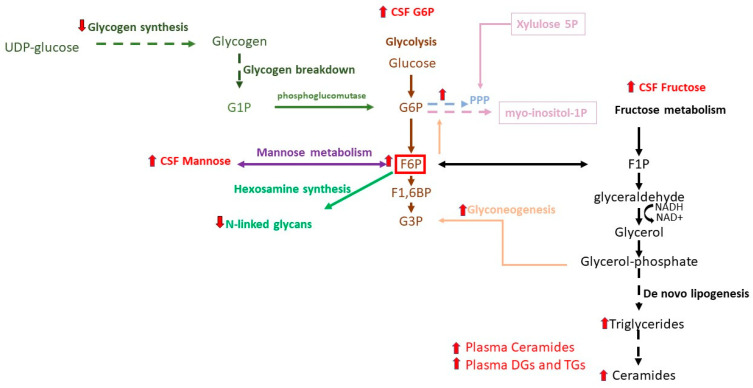
Schematic representation of interface of glycolysis with other metabolic pathways and altered metabolites in plasma and CSF. G6P, glucose 6-phosphate; F6P, fructose 6-phosphate; F1,6BP, fructose-1,6-bisphosphate; G3P, glyceraldehyde 3-phosphate; G1P, glucose 1-phosphate; F1P, fructose 1-phosphate; ADP, adenosine diphosphate; ATP, adenosine triphosphate; UDP-glucose, uridine diphosphate glucose; UDP-galactose, uridine diphosphate galactose; DHAP, dihydroxyacetone phosphate; GDP-mannose, guanosine diphosphate mannose.

**Figure 3 metabolites-15-00208-f003:**
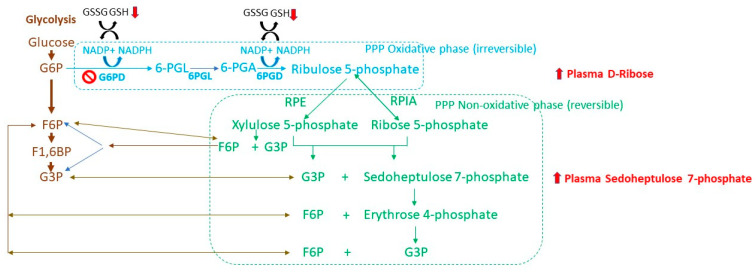
Schematic representation of oxidative (blue) and non-oxidative (green) phases of pentose phosphate pathway (PPP) and altered metabolites in plasma and CSF. G6P, glucose 6-phosphate; F6P, fructose 6-phosphate; F1,6BP, fructose-1,6-bisphosphate; G3P, glyceraldehyde 3-phosphate; G6PD, glucose-6-phosphate dehydrogenase; 6-PGL, 6-phosphogluconolactone; 6PGL, 6-phospho-gluconolactonase; 6-PGA, 6-phosphogluconate; 6PGD, 6-phosphogluconate dehydrogenase; RPE, ribulose-5-phosphate 3-epimerase; RPIA, ribose-5-phosphate isomerase A; NADPH, nicotinamide adenine dinucleotide phosphate.

**Figure 4 metabolites-15-00208-f004:**
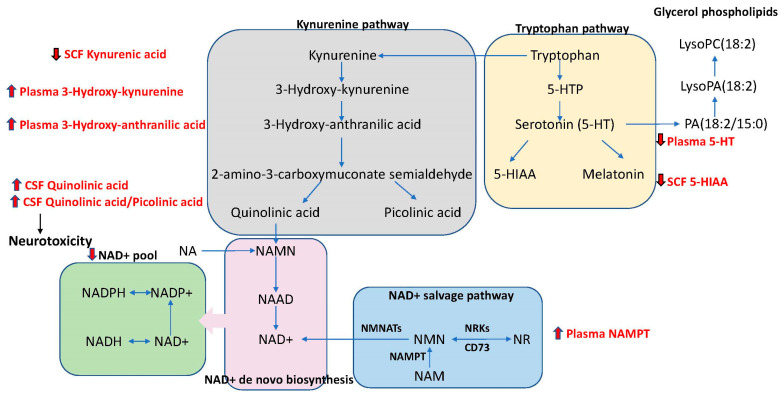
Schematic representation of NAD-Tryptophan-Kynurenine metabolic network and altered metabolites in plasma and CSF. 5-HTP, 5-Hydroxytryptophan; 5-HT, serotonin; 5-HIAA, 5-hydroxyindoleacetic acid; NAMN, nicotinamide mononucleotide; NAAD, nicotinic acid adenine dinucleotide; NMNATs, nicotinamide mononucleotide adenylyl transferases; NMN, nicotinamide mononucleotide; NAM, nicotinamide; NAMPT, nicotinamide phosphoribosyl transferase; NR, nicotinamide riboside; NRKs, nicotinamide riboside kinases; NADH, nicotinamide adenine dinucleotide; NADPH, nicotinamide adenine dinucleotide phosphate; NA, nicotinic Acid.

**Figure 5 metabolites-15-00208-f005:**
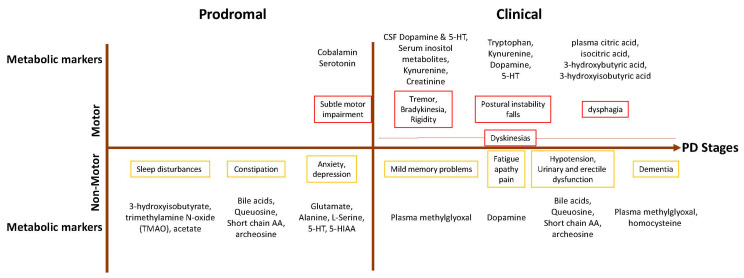
Schematic representation of Parkinson’s disease stages and metabolic markers: 5-HT, serotonin or 5-hydroxytryptamine; 5-HIAA, 5-hydroxyindoleacetic acid; CSF, cerebrospinal fluid.

**Table 1 metabolites-15-00208-t001:** Glycolysis, TCA, fructose, mannose, and redox pathway markers (*p* < 0.05).

Metabolite	PD Patients/Controls N	PD vs. Healthy Controls (Fold Change)	Biofluid	Method	References	Hoehn and Yahr (H-Y) Scale
Lactate	n = 101/n = 60	Increased (1.11)	CSF		[53]	2
Lactate	n=101/n=60	Increased (1.3)	CSF		[53]	3
Lactate	n = 20/n = 58	Increased (1.2)	CSF	NMR	[54]	1–4
Lactate	n = 10	Decreased (0.8)	CSF	NMR	[55]	2
Lactate	n = 65/n = 26	No change	CSF	HPLC	[56]	
Lactate	n = 151/n = 177	Increased (1.25)	plasma	UPLC-MS; LC-MS	[57]	
Lactate	n = 20/n = 20	Increased (1.1)	plasma	GC-MS	[58]	2
Lactate	n = 17/n = 22	Increased (2.25)	serum	NMR	[59]	2
Lactate/N-acetylaspartate	n = 14/n = 13	Increased (2.2)	The occipital lobe spectra	NMR	[51]	
Glucose	n = 20/n-58	No change	CSF	NMR	[54]	2
Glucose	n = 10	Decreased (0.8)	CSF	NMR	[55]	2
Glucose	n = 48/57	Decreased (0.6)	CSF	UHPLC-MS	[60]	
Glucose	n = 17/n = 22	Increased (2.6)	serum	NMR	[59]	
Glucose	n = 20/n = 20	Increased (1.1)	plasma	GC-MS	[58]	2
Pyruvate	n = 65/n = 26	No changes	CSF	HPLC	[56]	
Pyruvate	n = 20/n = 58	Decreased (0.8)	CSF		[54]	
Pyruvate	n = 151/n = 177	Increased (1.7)	plasma	UPLC-MS; LC-MS	[57]	
Pyruvate	n = 14/n = 65	No change	urine	GC-MS, LC-MS	[61]	1
Pyruvate	n = 59/n = 65	Increased (1.95)	urine	UPLC-MS; LC-MS	[61]	2–2.5
Pyruvate	n = 19/n = 65	Increased (1.6)	urine	GC-MS, LC-MS	[61]	3–4
Citrate	n = 17/n = 22	Increased (2.95)	serum	NMR	[59]	
Succinate	n = 34/n = 31	Increased (1.4)	plasma	NMR&MS	[62]	
Succinate	n = 50/n = 501	Decreased (0.9)	CSF	UHPLC-MS	[63]	
Succinate	n = 14/n = 65	Increased (1.36)	urine	GC-MS, LC-MS	[61]	1
Succinate	n = 59/n = 65	Increased (3.04)	urine	GC-MS; LC-MS	[61]	2–2.5
Succinate	n = 19/n = 65	Increased (2.1)	urine	GC-MS, LC-MS	[61]	3–4
Formic acid	n = 34/n = 31	Decreased (0.83)	plasma	NMR&MS	[62]	
Fructose	n = 44/n = 35	Increased (1.6)	CSF	GC-MS	[64]	DeNoPa cohort
Propionic acid	n = 20/n = 58	Decreased (0.25)	CSF	NMR	[54]	
Mannose	n = 44/n = 35	Increased (1.25)	CSF	GC-MS	[64]	DeNoPa cohort
Mannose	n = 10	Decreased (0.79)	CSF	NMR	[55]	2
Mannitol	n = 22/n = 28	Increased (1.32)	CSF	GC-MS, LC-MS	[65]	
Sorbitol	n = 22/n = 28	Increased (1.42)	CSF	GC-MS, LC-MS	[65]	
galactitol	n = 22/n = 28	Increased (1.26)	CSF	GC-MS, LC-MS	[65]	
Glycerol-3-phosphate	n = 22/n = 28	Increased (1.36)	CSF	GC-MS, LC-MS	[65]	
Complex I	n = 20/n = 17	Decreased (0.35)	Leukocytes		[66]	
Complex IV	n = 20/n = 17	Decreased (0.4)	Leukocytes		[66]	
Itaconate	n = 282/n = 185	Decreased (0.79)	serum	LC-HRMS	[67]	
Cysteine-S-Sulfate	n = 282/n = 185	Increased (1.56)	serum	LC-HRMS	[67]	
Cysteine-S-Sulfate	n = 28/n = 45	Increased (1.5)	serum	LC-MS	[68]	1–3
Cysteine-S-Sulfate	n = 8/n = 45	Increased (2)	serum	LC-MS	[68]	4

**Table 2 metabolites-15-00208-t002:** Tryptophan/kynurenine pathways.

Metabolite	PD Patients/ Controls N	PD vs. Healthy Controls (Fold Change)	Biofluid	Method	Reference	Hoehn and Yahr (H-Y) Scale
Tryptophan	n = 20/n = 20	Decreased (0.83)	CSF	GC-MS	[58]	
Tryptophan	n = 18/n = 7	Decreased (0.89)	Serum	UPLC-MS	[120]	2.9
Tryptophan betain	n = 18/n = 7	Decreased (0.4)	Serum	UPLC-MS	[120]	2.9
Serotonin (5-HT)	n = 82/n = 64	Decreased (0.62)	Plasma	HPLC-ECD	[127]	1.82
Hydroxyindoleacetic	n = 82/n = 64	Decreased (0.52)	Plasma	HPLC-ECD	[127]	1.82
Kynurenine	n = 14/n = 65	Increased (1.92)	urine	GC-MS, LC-MS	[61]	1
Kynurenine	n = 59/n = 65	Increased (3.91)	urine	GC-MS; LC-MS	[61]	2–2.5
Kynurenine	n = 19/n = 65	Increased (5.71)	urine	GC-MS, LC-MS	[61]	3–4

## Data Availability

No new data were created or analyzed in this study.

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
