# Peer review of "Metabolomics in Parkinson’s Disease and Correlation with Disease State"

_metabolites, 2025, doi:10.3390/metabo15030208_

Round 1
Reviewer 1 Report
Comments and Suggestions for Authors
The manuscript is about the key strength of metabolomics and its ability to provide a comprehensive snapshot of the metabolic state of an organism by analyzing metabolites in biofluids such as plasma, cerebrospinal fluid, urine, and saliva. This helps to identify specific metabolic changes associated with PD that may serve as potential biomarkers for early diagnosis and monitoring of disease progression. However, metabolomic analysis and data processing face several challenges.
Nomenclatures (e.g. FDG) need to be correct and consistent, and abbreviations need to be spelled out once or all.
The literature review is unsatisfactory and does not cover the related publications. The reviewer suggests including them, e.g. Shao et al. Mol. Neurodegeneration https://doi.org/10.1186/s13024-021-00425-8; Xin Li et al. Mol. Neurodegeneration https://doi.org/10.1007/s12035-021-02657-7.
The figures are of poor quality and the graphics are complicated.
The manuscript, and eventually the conclusion, needs to better explain the benefits and challenges of metabolomic analysis, which is a fascinating and promising area of research in Parkinson's disease (PD). Alterations in multiple metabolic pathways and their correlation with the progression of motor and non-motor features in PD provide valuable insights into the pathophysiology of the disease.
Standardization of results is also critical, as variations in sample collection, preparation, and analysis can lead to inconsistent results. Large cohort studies are needed to validate metabolomic profiles and ensure their reliability and reproducibility.
Metabolomics holds great potential for uncovering the underlying pathophysiological mechanisms of PD, and this manuscript should further elaborate on the identification of novel biomarkers. This will ultimately lead to the development of novel therapeutic strategies and improved patient management.
Author Response
The manuscript is about the key strength of metabolomics and its ability to provide a comprehensive snapshot of the metabolic state of an organism by analyzing metabolites in biofluids such as plasma, cerebrospinal fluid, urine, and saliva. This helps to identify specific metabolic changes associated with PD that may serve as potential biomarkers for early diagnosis and monitoring of disease progression. However, metabolomic analysis and data processing face several challenges.
We would like to thank the reviewers for their careful consideration and review of this manuscript. Attached, please find our responses. We have highlighted the corresponding revisions in the re-submitted file.
Nomenclatures (e.g. FDG) need to be correct and consistent, and abbreviations need to be spelled out once or all.
We thank the reviewer for this suggestion. We updated the above nomenclature (lines 101/104) as mentioned as well as for other abbreviations. We also added a full Abbreviations section at the end of the manuscript (lines 852-884).
The literature review is unsatisfactory and does not cover the related publications. The reviewer suggests including them, e.g. Shao et al. Mol. Neurodegeneration https://doi.org/10.1186/s13024-021-00425-8; Xin Li et al. Mol. Neurodegeneration https://doi.org/10.1007/s12035-021-02657-7.
Thank you for pointing out this oversight. We added the above references suggested as well as updated others to strengthen the supporting evidence in the article (lines 84-87, line 107, lines 137-141, lines 316-322, 341-344 and 351-354, lines 393-400, lines 416-422, and 423-427, lines 429-433, lines 466-472, lines 478-480, line 497-527, lines 685-696, lines 714-717, and lines 808-845).
The figures are of poor quality and the graphics are complicated.
We improved the quality of the figures by reworking all figures. We also simplified the pathways to reduce text. Further we added in information about changes in specific pathways seen in the literature to enhance the relevance to the text (Figures 1-4).
The manuscript, and eventually the conclusion, needs to better explain the benefits and challenges of metabolomic analysis, which is a fascinating and promising area of research in Parkinson's disease (PD). Alterations in multiple metabolic pathways and their correlation with the progression of motor and non-motor features in PD provide valuable insights into the pathophysiology of the disease.
We thank the reviewer for this excellent suggestion and have added several sections to the conclusion to highlight some of these challenges and potential benefits (lines 806-820 and lines 822-844).
Standardization of results is also critical, as variations in sample collection, preparation, and analysis can lead to inconsistent results. Large cohort studies are needed to validate metabolomic profiles and ensure their reliability and reproducibility.
This is an excellent point. We updated the manuscript to highlight some of the additional challenges in this area (lines 807-820). In lines 714-718 we also include some of the longitudinal clinical trials in which specific metabolites were measured. We also mention a couple studies with larger cohorts (line 255,257, lines 351-354).
Metabolomics holds great potential for uncovering the underlying pathophysiological mechanisms of PD, and this manuscript should further elaborate on the identification of novel biomarkers. This will ultimately lead to the development of novel therapeutic strategies and improved patient management.
Thank you for this point. We added further potential biomarker targets in lines 822-845 in the Conclusion section.

Reviewer 2 Report
Comments and Suggestions for Authors
The manuscript titled “Metabolomics in Parkinson’s Disease and correlation with disease state” has potential for publication in this journal. The review article is well-structured, and the authors have diligently compiled a wealth of data from recent studies. However, in its current form, the manuscript requires major revisions before it can be considered for publication. Several areas could benefit from additional detail or clarification to enhance the manuscript's utility for researchers and clinicians. Below are my comments and suggestions for improvement.
1) The manuscript discusses glycolysis, the TCA cycle, and pentose phosphate pathway alterations in PD. However, specific quantitative data comparing PD patients to healthy controls (e.g., fold changes in metabolites) would provide more insight. Could the authors include such metrics or elaborate on their implications?
2) The discussion on redox metabolites and their correlation with oxidative stress in PD is compelling. Can the authors detail the exact mechanisms by which these metabolites influence disease progression?
3) The authors describe metabolomics techniques (e.g., LC-MS/MS and GC-MS). It would be beneficial to specify which methods were predominant in generating the reviewed data. Were there any noted limitations in these methods that might affect the interpretation of results?
4) Identifying potential biomarkers is highlighted, but little is mentioned about their validation in large cohorts. Are there ongoing studies or data from clinical trials to support these findings?
5) The conclusion mentions the potential for therapeutic interventions. Could the authors hypothesize how metabolomics findings might translate into treatment strategies or preventive measures?
6) Although the manuscript references several figures, their connection to the discussion is somewhat limited. For example, Figure 1, depicting glucose metabolism, could be more effectively integrated to highlight the findings on glycolytic suppression in Parkinson's disease. Strengthening the link between the figures and the narrative would provide greater clarity and support for the key points discussed.
7) The citation style is consistent, but ensuring all key claims are adequately supported by references would strengthen the manuscript. Could additional studies be cited to provide further support for the less explored claims?
Author Response
We would like to thank the reviewers for their careful consideration and review of this manuscript. Attached, please find our responses. We have highlighted the corresponding revisions in the re-submitted file.
Point-by-point response to Comments and Suggestions for Authors
The manuscript titled “Metabolomics in Parkinson’s Disease and correlation with disease state” has potential for publication in this journal. The review article is well-structured, and the authors have diligently compiled a wealth of data from recent studies. However, in its current form, the manuscript requires major revisions before it can be considered for publication. Several areas could benefit from additional detail or clarification to enhance the manuscript's utility for researchers and clinicians. Below are my comments and suggestions for improvement.
1) The manuscript discusses glycolysis, the TCA cycle, and pentose phosphate pathway alterations in PD. However, specific quantitative data comparing PD patients to healthy controls (e.g., fold changes in metabolites) would provide more insight. Could the authors include such metrics or elaborate on their implications?
We appreciate this thoughtful comment. Specific metrics were added for studies in line 179/181, line 210-211, lines 231-233, lines 241-244, lines 248-252, lines 256-257, lines 267-269, lines 341-344, lines 351-354, lines 393-400, lines 446-455, lines 514-515. However, for some studies we could not obtain a specific fold change number due to the data presentation style in some articles. In addition, statistical values were added in several areas (e.g. lines 148-149, lines 181-182, lines 225-226, and 241-243).
2) The discussion on redox metabolites and their correlation with oxidative stress in PD is compelling. Can the authors detail the exact mechanisms by which these metabolites influence disease progression?
Thank you for pointing this out. To address this, we have further elaborated on metabolites of oxidative stress including xanthine oxidoreductase in lines 286-290 and added a mechanistic link between NADH imbalance and oxidative stress in lines 327-329. NADH imbalance was also addressed as a mechanism in lines 327-329 and Figure 1. We did already discuss decreased glutathione levels in lines 264-274 and the increase in nicotinamide levels in lines 316-317 as mechanisms for increased oxidative stress in PD.
3) The authors describe metabolomics techniques (e.g., LC-MS/MS and GC-MS). It would be beneficial to specify which methods were predominant in generating the reviewed data. Were there any noted limitations in these methods that might affect the interpretation of results?
In order to address this point, we added reference 17 which described which techniques (GC/MS vs LC/MS) were utilized in PD-related studies (see Lines 78-80). We added a paragraph to discuss potential limitations including metabolite reproducibility in Conclusions (lines 808-820).
4) Identifying potential biomarkers is highlighted, but little is mentioned about their validation in large cohorts. Are there ongoing studies or data from clinical trials to support these findings?
We agree with the reviewer’s comment. We added some of the longitudinal trials in which testing for certain metabolites has occurred (lines 714-717). Additionally, we added references for a couple studies which have large cohorts, although these were sparse (lines 255, 257, lines 351-354).
5) The conclusion mentions the potential for therapeutic interventions. Could the authors hypothesize how metabolomics findings might translate into treatment strategies or preventive measures?
Thank you for bringing up this important point. To address this, we added several examples in the conclusions section lines 822-845.
6) Although the manuscript references several figures, their connection to the discussion is somewhat limited. For example, Figure 1, depicting glucose metabolism, could be more effectively integrated to highlight the findings on glycolytic suppression in Parkinson's disease. Strengthening the link between the figures and the narrative would provide greater clarity and support for the key points discussed.
Thank you for the excellent suggestion. We added in information about changes in specific pathways seen in the literature to enhance the relevance to the text (Figures 1-4). We also simplified the pathway depiction.
7) The citation style is consistent, but ensuring all key claims are adequately supported by references would strengthen the manuscript. Could additional studies be cited to provide further support for the less explored claims?
Thank you for pointing out this oversight. We updated references to strengthen the supporting evidence in the article in multiple places (lines 84-87, lines 106-107, lines 137-141, lines 316-322, 341-344 and 351-354, lines 393-400, lines 416-422, and 423-427, lines 429-433, lines 466-472, lines 478-480, line 497-527, lines 685-696, lines 714-717, and lines 808-845).

Round 2
Reviewer 2 Report
Comments and Suggestions for Authors
Dear Editor,
Thank you for the opportunity to review this manuscript. I sincerely appreciate the trust you have placed in me. Please find my review comments below.
The submitted manuscript entitled “Metabolomics in Parkinson’s Disease and correlation with disease state” seems potential publication in this journal. The article is well-structured and effectively presented. The manuscript has improved significantly, the authors have made significant improvements in addressing the initial review comments. While they have incorporated additional details and clarifications, a few areas such as explicit figure integration and a structured therapeutic roadmap could still be refined further. Overall, minor revisions are recommended to ensure maximal clarity and readability.
- Some sections, particularly those discussing metabolic pathways, contain dense technical details. Consider simplifying key concepts or incorporating summary tables to compare major findings will be helpful for the readers for better understating.
- Ensure consistency in referring to metabolic pathways and metabolites. For example, "TCA cycle" and "Krebs cycle" are used interchangeably; it may be helpful to clarify terminology at the beginning.
- While figures are referenced throughout the manuscript, their connection to the discussion remains limited. For instance, Figure 1 on glucose metabolism could be more explicitly linked to the section on glycolytic suppression in PD. The authors should ensure that each figure is directly referenced and contextualized within the relevant discussion.
- Please explain Could metabolomics lead to targeted interventions, such as dietary modifications or metabolic therapy? What are the major hurdles in translating these findings into clinical applications?
Author Response
Point-by-point response to Comments and Suggestions for Authors
We greatly appreciate the time taken by the reviewers and their consideration and review. We appreciate the suggestions, which have helped improve our manuscript. Please find our responses below. All new changes in the manuscript are highlighted in blue (prior Round 1 changes are in yellow).
The submitted manuscript entitled “Metabolomics in Parkinson’s Disease and correlation with disease state” seems potential publication in this journal. The article is well-structured and effectively presented. The manuscript has improved significantly, the authors have made significant improvements in addressing the initial review comments. While they have incorporated additional details and clarifications, a few areas such as explicit figure integration and a structured therapeutic roadmap could still be refined further. Overall, minor revisions are recommended to ensure maximal clarity and readability.
1. Some sections, particularly those discussing metabolic pathways, contain dense technical details. Consider simplifying key concepts or incorporating summary tables to compare major findings will be helpful for the readers for better understating.
We thank the reviewer for this comment which lead us to include two additional tables-- Table 1 (pages 16-17) and Table 2 (page 17-18)--to help clarify some of these findings. We also removed some of the technical details in the text as they are now included in table form.
2. Ensure consistency in referring to metabolic pathways and metabolites. For example, "TCA cycle" and "Krebs cycle" are used interchangeably; it may be helpful to clarify terminology at the beginning.
Thank you for pointing this out. We removed the mention to Krebs cycle in line 128 and made sure that only the term “tricarboxylic acid” or “TCA” is referenced throughout the manuscript.
3. While figures are referenced throughout the manuscript, their connection to the discussion remains limited. For instance, Figure 1 on glucose metabolism could be more explicitly linked to the section on glycolytic suppression in PD. The authors should ensure that each figure is directly referenced and contextualized within the relevant discussion.
We apologize for misinterpretation of the Round 1 comment on this issue. To connect the figures and tables more directly to the text, we added additional references throughout the text of the manuscript (Figure 1: lines 132, 135, 141, 155, 163, 232, 304; Figure 2: line 177, Figure 3: 189, 195, 203; Figure 4: 342, 352, 354, 342, 374; Figure 5: lines 494, 508, 587, 616, 647, 699; Table 1 (139-140, 155, 182, 231, 240, 342, 352, 354; and Table 2 line 339, 349, 339, and 349).
4. Please explain Could metabolomics lead to targeted interventions, such as dietary modifications or metabolic therapy? What are the major hurdles in translating these findings into clinical applications?
This is a good point and we added some examples of dietary modifications that could be utilized and have been suggested in the literature (lines 851-859). Overall, this is a limited summarization of ongoing work, as a thorough discussion would be out of range of this review article. A discussion of limitations including standardization in collection and cohort testing was added in prior round to the Discussion section (lines 812-824).
